# Real-Time Calibration of Magnetometers Using the RLS/ML Algorithm

**DOI:** 10.3390/s20020535

**Published:** 2020-01-18

**Authors:** Guocan Cao, Xiang Xu, Dacheng Xu

**Affiliations:** School of Electronic and Information Engineering, Soochow University, Suzhou 215006, China; 20175228034@stu.suda.edu.cn (G.C.); hsianghsu@163.com (X.X.)

**Keywords:** magnetometer calibration, real-time, recursive least square estimation, maximum likelihood estimation

## Abstract

This study presents a new real-time calibration algorithm for three-axis magnetometers by combining the recursive least square (RLS) estimation and maximum likelihood (ML) estimation methods. Magnetometers are widely employed to determine the heading information by sensing the magnetic field of earth; however, they are vulnerable to ambient magnetic disturbances. This makes the calibration of a magnetometer inevitable before it is employed. In this paper, first, a complete measurement error model of the magnetometer is studied, and a simplified model is developed. Then, the real-time RLS algorithm is introduced and discussed in detail, and the unbiased optimal ML is utilized to improve the accuracy of the parameter estimation. The proposed algorithm is advantageous in correcting the parameters in real time and simultaneously obtaining unbiased parameter estimation. Finally, the simulation and experimental results demonstrate that both the accuracy and computational speed of the proposed algorithm is better than those of the widely used bath-processing method. Moreover, the proposed calibration method can be adopted for calibrating other three-axis sensors.

## 1. Introduction

With the rapid development of micro electromechanical system (MEMS) technology, developing accurate long-term positioning systems based on the MEMS inertial devices without any external signals may become possible. Currently, most magnetic and inertial measurement units (MIMU) based on the MEMS technology integrate three accelerometers, gyroscopes, and magnetometers each, assembled in three orthogonal directions. Due to the weak yaw observation of the conventional navigation systems, magnetometers are widely utilized for heading estimation based on the principle that the local magnetic field points to the north [1,2,3,4].

Nevertheless, the yaw information obtained by the magnetic sensor is usually vulnerable to magnetic field disturbances [5]. Further, the result obtained is sometimes worse with magnetometers than those without magnetometers, if the local magnetic field is contaminated. The magnetic disturbances can be classified into hard iron and soft iron disturbances. The hard iron disturbances are caused by ferromagnetic materials with a permanent magnetic field, e.g., magnets and speakers, which are time-invariant, whereas the soft iron disturbances are deflections or alterations in the existing magnetic field, which are caused by magnetized materials such as steel shield and batteries [2]. Moreover, magnetometers inherently possess non-orthogonality, scaling, and bias errors. Thus, magnetometer calibration is indispensable in realizing accurate orientation estimation.

The conventional magnetometer calibration methods assume that a reference sensor, which can provide accurate heading information, is available [6,7,8]. A well-known example of such an approach is compass swinging [7]. However, this method requires external equipment for assistance, making it expensive and impractical for a wide range of applications. Thus, magnetometer calibration using the information provided only by the MIMU has been increasingly popular. Magnetometer calibration without the aid of external equipment can generally be grouped to two categories: one that uses the data from only the magnetometer, and the other that performs calibration along with inertial sensors. In the first category, a classical method is used to formulate the calibration problem as an ellipsoid fitting problem based on the least-square (LS) method, similar to mapping an ellipsoid data to a sphere [9,10]. This approach has the advantage that solving an ellipsoid fitting problem is simple. However, it has some clear drawbacks: sufficient measured data must be available to avoid the emergence of singular matrix for the least-squares solution; determining the required volume of measured data is difficult. Another class is the TWOSTEP batch method designed by Gebre-Egziabher et al. and Zhang et al. [11,12], which is based on the differences between the actual and the measured unit vectors. In this method, first, the centering approximation is utilized to yield a good initial estimation based on the LS method. Second, the non-orthogonality and the scaling and bias errors are estimated using the bath Gauss-Newton method. However, the estimation result is not an optimal solution because of the existence of noise in the least square coefficient matrix. Furthermore, the magnetometer calibration problem can be formulated as a suboptimal maximum likelihood estimation (called NM) by considering the smallest parameter dimension [13,14]. This derivation is based on the fact that the magnitude of the calibrated measurements must be constant in a homogeneous magnetic field; further, the Gauss–Newton method is utilized to address the NM problem. However, this method has the following limitations: first, this method yields biased results because of the quadratic item of the noise, which has an imperative effect on the estimation. Second, the objective function of the NM is quartic in parameters, which complicates the calibration with multiple minima and maxima, thus necessitating a good estimation for the nonlinear solver. Wu et al. [15] proposed the use of quadratic unbiased optimal maximum likelihood (ML) estimation, which is superior to the conventional quartic suboptimal ML estimation both in accuracy and stability. It must be noted that all the above-mentioned methods are batch-processing methods. Many studies in the literature propose a real-time calibration algorithm for magnetometers, specifically the filtering algorithm, to address the problems in the above-mentioned methods [16,17]. Although the real-time calibration algorithm can estimate the parameters in real time and reduce computational loads, the filtering algorithm is still a biased estimation.

In the second category, the magnetometer calibration is performed with assistance from inertial sensors [18,19,20,21,22,23]. This approach has the advantage that it can perform an alignment estimation of the sensor axes between the inertial sensors and the magnetometer. However, it increases the complexity of the system and may introduce unnecessary errors from the inertial sensors (gyroscope/accelerometer). In actual inertial navigation systems, the alignment error between the sensor axes can be ignored by comparison with the other errors. Thus, magnetometer calibration using only the information itself is the main objective of this research.

To address the above-mentioned problems, this research proposes a combination of the recursive least square (RLS) estimation method and the optimal ML estimation algorithm to successfully realize magnetometer calibration utilizing only the magnetometer information. The proposed algorithm can estimate the parameters in real time and simultaneously obtain unbiased optimal estimation results. Moreover, this algorithm can adaptively detect the calibration implementation, which is discussed in detail in the following section. Notably, because of its similar measurement model, the proposed calibration algorithm can be applied to other three-axis sensors.

The following sections are organized as given below. In Section 2, a detailed description of our proposed RLS algorithm is presented. In Section 3, the quadratic optimal ML estimation is discussed in detail to further improve the accuracy. In Section 4, the results of numerous simulations and experiments performed in this study are provided and discussed. Finally, the conclusions are provided in Section 5.

## 2. Sensor Model and Initial Estimation

### 2.1. Sensor Model

Based on previous studies [13,18,24], the complete error model of the magnetometer is given by,
(1)m˜b=SMCNO(CSICnbmn+bHI)+bM+nm=‖mn‖SMCNOCSICnbm¯n+SMCNObHI+bM+nm=PCnbm¯n+hm+nm
where P=mnSMCNOCSI, hm=SMCNObHI+bM=[h1h2h3]T, and m¯n=mn/‖mn‖. m˜b=[m˜xb m˜yb m˜zb]T∈ℝ3 denotes the measured magnetic field vector, nm∈ℝ3 denotes the zero-mean Gaussian noise, mn∈ℝ3 is the local geomagnetic field vector in the navigational frame, whose magnitude is constant when the external magnetic field does not change over time, and m¯n is the unit vector obtained by normalizing mn. The attitude matrix Cnb∈SO(3) transforms the magnetic field vector from the navigational frame (n-frame) to the body frame (b-frame), where SO(3) denotes a set of 3×3 orthogonal matrices. SM∈D(3) represents the scaling matrix, where D(3) denotes a set of 3×3 diagonal matrices and CNO is a non-orthogonal error matrix. mSI is given as mSI=CSICnbmn, where CSI∈ℝ3×3 is the soft iron transformation matrix. mSI and bHI denote the soft and hard iron effects, respectively, and bM is the null-shift of the magnetometer. In this paper, the navigational frame is defined as East-North-Up and the body frame is defined as Right-Forward-Up.

Clearly, the purpose of the three-axis magnetometer calibration is to obtain the optimal parameters P and $h_{m}$. We assume $P^{-1}\;=\;\mathrm{QR}$ based on the orthogonal-triangular (QR) decom-position, where Q and R denote the orthogonal and upper triangular matrices with positive diagonal entries [15]. The first item in Equation (1) can be written as,
(2)$$PC_{n}^{b}{\bar{m}}^{n}=R^{-1}Q^{T}C_{n}^{b}{\bar{m}}^{n}=Tm^{b^{\prime }},$$
where ${T=R}^{-1}$, which is an upper triangular matrix, and $m^{b^{\prime }}=Q^{T}C_{n}^{b}{\bar{m}}^{n}$, which is a unit vector. Finally, the magnetometer sensor model can be expressed as,
(3)$${\tilde{m}}^{b}=Tm^{b^{\prime }}+h_{m}+n_{m},$$
and the calibrated magnetometer measurement can be expressed as,
(4)$$m^{b^{\prime }}=T^{-1}({\tilde{m}}^{b}-h_{m}).$$

### 2.2. Proposed Initial Estimation Method

The most popular method to achieve a good initial estimation is the method based on the batch-processing LS [15,25,26]. This method requires a batch of data, which cannot correct the parameters in real time to improve the accuracy.

In this paper, a real-time initial estimation method based on the RLS is proposed. With the increase in the number of iterations, the state vector tends to converge, which indicates a good initial estimation. Moreover, to address the above-mentioned drawbacks of the LS method, the RLS can develop a real-time parameter estimation, which is implemented once the state covariance matrix converges well.

By performing a transformation and a modulus operation, Equation (3) can be written as,
(5)$$\begin{array}{lll} {\left\|m^{b^{\prime }}\right\|}^{2} & = & \|R\left({\tilde{m}}^{b}-h_{m}-n_{m}\right){\|}^{2}\\ & = & {\left\|R\left({\tilde{m}}^{b}-h_{m}\right)\right\|}^{2}-2{\left({\tilde{m}}^{b}-h_{m}\right)}^{T}R^{T}Rn_{m}+n_{m}^{T}R^{T}Rn_{m}\\ & = & {\left\|R\left({\tilde{m}}^{b}-h_{m}\right)\right\|}^{2}+v \end{array}$$
where $v=-2{\left({\tilde{m}}^{b}-h_{m}\right)}^{T}R^{T}Rn_{m}+n_{m}^{T}R^{T}Rn_{m}$, which is not the exact Gaussian noise because it contains a quadratic item of $n_{m}$. By expanding Equation (5), the following equation is obtained:(6)$$\begin{array}{ccc} z & = & \mathrm{Hx}\;+\;v \end{array},$$
where $H=[{\tilde{m}}_{y}^{b^{2}}\;{\tilde{m}}_{z}^{b^{2}}\;{\tilde{m}}_{x}^{b}\cdot {\tilde{m}}_{y}^{b}\;{\tilde{m}}_{x}^{b}\cdot {\tilde{m}}_{z}^{b}\;{\tilde{m}}_{y}^{b}\cdot {\tilde{m}}_{z}^{b}\;-{\tilde{m}}_{x}^{b}\;-{\tilde{m}}_{y}^{b}\;-{\tilde{m}}_{z}^{b}\;1]$, $z=-{\tilde{m}}_{x}^{b^{2}}$. Assuming ${A=R}^{T}R$, which is a symmetric matrix, the state vector x can be expressed as,
(7)$$\begin{array}{lll} x & = & {\left[\begin{array}{ccc} \begin{array}{ccc} \frac{a_{22}}{a_{11}} & \frac{a_{33}}{a_{11}} & \frac{{2a}_{12}}{a_{11}} \end{array} & \begin{array}{ccc} \frac{{2a}_{13}}{a_{11}} & \frac{{2a}_{23}}{a_{11}} & k_{1} \end{array} & \begin{array}{ccc} k_{2} & k_{3} & k_{4} \end{array} \end{array}\right]}^{T}\\ \; & = & {\left[\begin{array}{ccc} \begin{array}{ccc} x_{1} & x_{2} & x_{3} \end{array} & \begin{array}{ccc} x_{4} & x_{5} & x_{6} \end{array} & \begin{array}{ccc} x_{7} & x_{8} & x_{9} \end{array} \end{array}\right]}^{T},\; \end{array}$$
where $a_{\mathrm{ij}}\left(i,j=1,\;2,\;3\right)$ denotes the element in the ith row and jth column of matrix A, $x_{i}\left(i=1,\;2,\;\cdots ,\;9\right)$ denotes the ith element of state vector x, and $k_{i}\left(i\;=\;1,\;2,\;3,\;4\right)$ denotes the expression consisting of $a_{\mathrm{ij}}$ and $h_{i}\left(i=1,\;2,\;3\right)$. $k_{i}\left(i\;=\;1,\;2,\;3,\;4\right)$ can be written as,
(8)$$\{\begin{array}{l} k_{1}=2h_{1}+2\frac{a_{12}}{a_{11}}h_{2}+2\frac{a_{13}}{a_{11}}h_{3}\\ k_{2}=2\frac{a_{12}}{a_{11}}h_{1}+2\frac{a_{22}}{a_{11}}h_{2}+2\frac{a_{23}}{a_{11}}h_{3}\\ k_{3}=2\frac{a_{13}}{a_{11}}h_{1}+2\frac{a_{23}}{a_{11}}h_{2}+2\frac{a_{33}}{a_{11}}h_{3}\\ k_{4}=h_{1}^{2}+\frac{a_{22}}{a_{11}}h_{2}^{2}+\frac{a_{33}}{a_{11}}h_{3}^{2}+2\frac{a_{12}}{a_{11}}h_{1}h_{2}+2\frac{a_{23}}{a_{11}}h_{2}h_{3}+2\frac{a_{13}}{a_{11}}h_{1}h_{3}-\frac{1}{a_{11}}{\left\|m^{b^{\prime }}\right\|}^{2} \end{array}$$

Then, the RLS method is utilized to update the state vector x. The updated equation is as follows:(9)$$\{\begin{array}{l} K=P_{k-1}H_{k}^{T}{\left(H_{k}P_{k-1}H_{k}^{T}+\delta_{k}\right)}^{-1}\\ {\hat{x}}_{k}={\hat{x}}_{k-1}+K\left(z_{k}-H_{k}{\hat{x}}_{k-1}\right)\\ P_{k}=\left(I_{k}-KH_{k}\right)P_{k-1} \end{array}$$
where K is the gain matrix, $\delta_{k}$ is the covariance value for measurement noise, and $I_{k}$ is a unit matrix of dimension 9 × 9. Remarkably, the diagonal elements of P tend to be stable when the state vector is convergent, based on the basic theory of filter. Therefore, by comparing the defined parameter s with an experienced threshold associated with the used sensor, we can detect whether the state vector has converged, as follows:(10)$$\begin{array}{ccc} s & = & \sum_{i=1}^{9}\sqrt{P_{\mathrm{ii}}}<\gamma \;i\;=\;1,\;2,\;\cdots ,\;9 \end{array}.$$

It must be noted that the experienced threshold γ in this paper can be obtained by experiments, which is introduced in Section 4.

Using Equations (7) and (8), ${\hat{a}}_{ij}$ and ${\hat{h}}_{i}$ can be obtained from the iteration estimation results of the state vector, as follows:(11)$$\{\begin{array}{l} {\hat{h}}_{2}=\frac{\left({\hat{x}}_{4}{\hat{x}}_{6}-2{\hat{x}}_{8}\right)\left({\hat{x}}_{3}{\hat{x}}_{4}-2{\hat{x}}_{5}\right)-\left({\hat{x}}_{3}{\hat{x}}_{6}-2{\hat{x}}_{7}\right)\left({\hat{x}}_{4}^{2}-4{\hat{x}}_{2}\right)}{{\left({\hat{x}}_{3}{\hat{x}}_{4}-2{\hat{x}}_{5}\right)}^{2}-\left({\hat{x}}_{4}^{2}-4{\hat{x}}_{2}\right)\left({\hat{x}}_{3}^{2}-4{\hat{x}}_{1}\right)}\\ {\hat{h}}_{3}=\frac{\left({\hat{x}}_{3}{\hat{x}}_{6}-2{\hat{x}}_{7}\right)\left({\hat{x}}_{3}^{2}-4{\hat{x}}_{1}\right){\hat{h}}_{2}}{\left({\hat{x}}_{3}{\hat{x}}_{4}-2{\hat{x}}_{5}\right)}\\ {\hat{h}}_{1}=\frac{{\hat{x}}_{4}{\hat{x}}_{6}-{\hat{x}}_{4}{\hat{h}}_{2}-{\hat{x}}_{4}^{2}{\hat{h}}_{3}}{2{\hat{x}}_{4}}\\ {\hat{a}}_{11}=\frac{{\left\|m^{b^{\prime }}\right\|}^{2}}{{\hat{h}}_{1}^{2}+{\hat{x}}_{1}{\hat{h}}_{2}^{2}+{\hat{x}}_{2}{\hat{h}}_{3}^{2}+{\hat{x}}_{3}{\hat{h}}_{1}{\hat{h}}_{2}+{\hat{x}}_{4}{\hat{h}}_{1}{\hat{h}}_{3}+{\hat{x}}_{5}{\hat{h}}_{2}{\hat{h}}_{3}-+{\hat{x}}_{9}}\\ {\hat{a}}_{22}={\hat{a}}_{11}{\hat{x}}_{1}\\ {\hat{a}}_{33}={\hat{a}}_{11}{\hat{x}}_{2}\\ {\hat{a}}_{12}=\frac{{\hat{a}}_{11}{\hat{x}}_{3}}{2}\\ {\hat{a}}_{13}=\frac{{\hat{a}}_{11}{\hat{x}}_{4}}{2}\\ {\hat{a}}_{23}=\frac{{\hat{a}}_{11}{\hat{x}}_{5}}{2} \end{array}$$
where the hatted quantities denote the final parameters of the estimated results; thus, $\hat{A}$ and $\hat{h}$ can be obtained. Further, $\hat{R}=chol\left(\hat{A}\right)$, where chol(·) denotes the matrix Cholesky factorization.

Using the proposed RLS, the state vector and the state covariance matrix converge well to produce a good estimation result.

## 3. Performing Magnetometer Calibration

Although a good initial estimation can be obtained using the RLS method, it is important to note that this method is a biased estimation method because it contains a quadratic item of the noise in Equation (5). Thus, a quadratic optimal ML estimation is adopted to address the above-mentioned problems, which can be regarded unbiased estimations. The objective function can be described as,
(12)$$f=\underset{\theta_{ml}}{\min}{\sum_{k=1}^{N}\left\|{\tilde{m}}_{k}^{b}-Tm_{k}^{b^{\prime }}-h_{m}\right\|}^{2}+\lambda_{k}\left({\left\|m_{k}^{b^{\prime }}\right\|}^{2}-1\right),$$
with variables $\theta_{\mathrm{ml}}=\left\{{T,\;h}_{m},\;m_{k}^{b^{\prime }},\;\lambda_{k}\right\}$, where $\lambda_{k}$ is the Lagrangian coefficient for the norm constraint $m_{k}^{b^{\prime }}$. Due to its good initial estimation characteristic, the Gauss–Newton method is adopted in this paper. The updated equation can be written as,
(13)$$\begin{array}{ccc} x_{k+1} & = & x_{k}-{\left[\nabla^{2}{f|}_{x}\right]}^{-1}\nabla {f|}_{x} \end{array},$$
where ∇f and $\nabla^{2}f$ denote the Jacobian vector and Hessian matrix, respectively. The state vector is given by $\begin{array}{ccc} x & = & \left[\begin{array}{ccc} \begin{array}{ccc} {\mathrm{vec}}^{T}\left(T\right) & h^{T} & m_{1}^{b^{\prime }} \end{array} & \begin{array}{ccc} \cdots & m_{N}^{b^{\prime }} & \lambda_{1} \end{array} & \begin{array}{cc} \cdots & \lambda_{N} \end{array} \end{array}\right] \end{array}$, and the Jacobian vector by,
(14)$$\begin{array}{ccc} \nabla {\;f|}_{x} & = & \left[\begin{array}{ccc} \nabla {\;f|}_{T}^{T} & \nabla {\;f|}_{h}^{T} & \begin{array}{cc} \nabla {\;f|}_{m_{k}^{b^{\prime }}}^{T} & \nabla {\;f|}_{\lambda_{k}}^{T} \end{array} \end{array}\right] \end{array}\;k=1,2,\cdots ,N.$$

Let $V_{k}={\tilde{m}}_{k}^{b}-Tm_{k}^{b}-h_{m}$; then
(15)$$\begin{array}{ccc} \nabla {\;f|}_{T}^{T} & = & -2\sum_{k=1}^{N}m_{k}^{b^{\prime }}⊗v_{k} \end{array},\;\begin{array}{ccc} \nabla {\;f|}_{h}^{T} & = & -2\sum_{k=1}^{N}v_{k} \end{array},\;\begin{array}{ccc} \nabla {\;f|}_{m_{k}^{b^{\prime }}}^{T} & = & {-2T}^{T}v_{k}{+2\lambda }_{k}m_{k}^{b^{\prime }} \end{array},\;\begin{array}{ccc} \nabla {\;f|}_{\lambda_{k}}^{T} & = & \|m_{k}^{b^{\prime }}{\|}^{2}-1 \end{array}.$$

In addition, the Hessian matrix becomes,
(16)$$\nabla^{2}f^{|}{}_{x}=\left[\begin{array}{cccccc} H_{TT} & H_{Th} & H_{Tm_{k}^{b^{\prime }}} & \cdots & 0_{9\times 1} & \cdots \\ H_{Th}^{T} & H_{hh} & H_{hm_{k}^{b^{\prime }}} & \cdots & 0_{3\times 1} & \cdots \\ H_{Tm_{k}^{b^{\prime }}}^{T} & H_{hm_{k}^{b^{\prime }}}^{T} & H_{m_{k}^{b^{\prime }}m_{k}^{b^{\prime }}} & \cdots & H_{\lambda_{k}m_{k}^{b^{\prime }}} & \cdots \\ \vdots & \vdots & \vdots & \ddots & \vdots & \ddots \\ 0_{9\times 1}^{T} & 0_{3\times 1}^{T} & H_{\lambda_{k}m_{k}^{b^{\prime }}}^{T} & \cdots & 0 & \cdots \\ \vdots & \vdots & \vdots & \ddots & \vdots & \end{array}\right].$$

Further,
(17)$$H_{\mathrm{TT}}\;=\;2\sum_{k=1}^{N}\left(m_{k}^{b^{\prime }}m_{k}^{b^{\prime }}{}^{T}\right)⊗I_{3},\;H_{\mathrm{Th}}\;=\;2\sum_{k=1}^{N}m_{k}^{b^{\prime }}⊗I_{3},\;H_{\mathrm{hh}\;}=2{NI}_{3},\;H_{Tm_{k}^{b^{\prime }}}=2\left(\left(m_{k}^{b^{\prime }}⊗I_{3}\right)T-I_{3}⊗v_{k}\right),\;H_{m_{k}^{b^{\prime }}m_{k}^{b^{\prime }}}=2T^{T}T+2\lambda_{k}I_{3},\;H_{hm_{k}^{b^{\prime }}}=2T,\;H_{\lambda_{k}m_{k}^{b^{\prime }}}=2m_{k}^{b^{\prime }},$$
where ⊗ denotes the Kronecker product, and A⊗B denotes the Kronecker product of A and B excluding the corresponding lower triangular rows and columns, i.e., the 2th, 3th, and 6th rows and columns are removed from the result of A⊗B. For an optimal ML estimation, $T_{\mathrm{init}}=R_{\mathrm{init}}^{-1}$, which can be obtained by the initial estimation; the initial Lagrangian coefficient $\lambda_{\mathrm{init}}=0$, and the initial $m_{k}^{b^{\prime }}$ can be obtained from Equation (4).

## 4. Simulation and Experimental Results

### 4.1. Simulation Results

The performed simulations and their results are provided in this section to study the performance of the proposed algorithm. The parameters such as the geomagnetic field unit vector ${\bar{m}}_{n}={\left[\begin{array}{ccc} -0.0695 & 0.6720 & -0.7373 \end{array}\right]}^{T}$ in Suzhou city, and P and $h_{m}$, used in the simulation for the measurement model (1) are given by,
(18)$$\begin{array}{ccc} P & = & \left[\begin{array}{ccc} 0.7 & -0.8 & 0.4\\ 1.1 & 0.3 & -0.1\\ -0.3 & 0.6 & 0.7 \end{array}\right] \end{array}\begin{array}{ccc} h_{m} & = & \left[\begin{array}{c} 0.5\\ 1.7\\ 2.6 \end{array}\right] \end{array}.$$

Moreover, the standard deviation of the measurement noise is σ=0.003; the attitude matrix $C_{n}^{b}$ can be expressed as,
(19)$$C_{n}^{b}=\left[\begin{array}{ccc} \cos\phi \cos\psi -sin\phi \sin\theta \sin\psi & \cos\phi \sin\psi +sin\phi \sin\theta \cos\psi & -sin\phi \cos\theta \\ -\cos\theta \sin\psi & \cos\theta \cos\psi & \sin\theta \\ sin\phi \cos\psi +\cos\phi \sin\theta \sin\psi & sin\phi \sin\psi -\cos\phi \sin\theta \cos\psi & \cos\phi \cos\theta \end{array}\right],$$
where ϕ, θ, and ψ denote the roll, pitch, and yaw angles, respectively, and are taken as,
(20)$$\{\begin{array}{l} \begin{array}{ccc} \phi & = & 20^{{}^{\circ}}\sin\left(20\pi k/N+\pi /2\right) \end{array}\\ \begin{array}{ccc} \theta & = & 20^{{}^{\circ}}\sin\left(20\pi k/N\right) \end{array}\\ \begin{array}{ccc} \psi & = & 360^{{}^{\circ}}k/N \end{array}, \end{array}$$
where k = 1, 2, ⋯, N.

The ellipsoid and sphere fitting results obtained from 300 simulations of the measurement model (1) are shown in Figure 1. Moreover, the changes in the state vector x with or without noise are plotted in Figure 2. The initial state vector $x_{0}\;=\;{\left[\begin{array}{ccccccccc} 1 & 1 & 0 & 0 & 0 & 0 & 0 & 0 & -1 \end{array}\right]}^{T}$ by assuming zero external magnetic field interference, i.e., A is assumed as a unit matrix; $h_{m}$ is the zero vector in Equation (7). The initial covariance matrix for the state vector $P_{0}$ must be set sufficiently large to achieve a better estimation. If $P_{0}$ is very small, the state vector may converge well earlier. Thus, an optimal estimation may not be achieved in the following recursive algorithm. In this paper, the value is selected as $P_{0}=diag\left(\left[\begin{array}{c} \begin{array}{ccccccccc} 1 & 1 & 2 & 2 & 2 & 6 & 6 & 6 & 1 \end{array} \end{array}\right]\times 10,000\right)$. In Figure 2, the red, blue, and green lines represent the order of parameters, and the dotted line represents the real value. For example, in the top image in Figure 2a, the red, blue, and green lines denote X1, X2, and X3, respectively. A comparison of Figure 2a,b indicates that the state vector without noise is almost equal to the real vector calculated using the given parameters, whereas the state vector with noise is not equal to the real vector. This result clearly demonstrates that the RLS method is a biased estimation method.

Figure 3a plots the changes in the square root of the diagonal elements of the covariance matrix $\sqrt{P_{ii}}$ with noise, which clearly converges well after 200 iterative operations. Thus, the parameter s at the 200th iteration can be considered the threshold γ to detect if the state vector has converged. To obtain γ sufficiently and reasonably, 80 Monte Carlo (MC) runs were performed. The variation curves of the MC runs were concurrent, as shown in Figure 3b. Further, this result agreed well with the expected results. Finally, the threshold is selected as the mean for all the 80 MC simulations.

Figure 4 plots the magnitudes of all the data points before and after the application of the iterative calibrated algorithm. The convergent point in the proposed estimation method is ${\hat{x}}_{195}$, which was used to calculate the magnitudes of all the data points. As shown in Figure 4, the magnitude of the data points of the proposed algorithm wavers near 1 with a small error, which corresponds well with the theory above (shown in Equation (4)). Moreover, it must be noted that although the later 105 data points were not used in the calibration, their magnitudes were steady, which clearly demonstrates the superiority of the proposed algorithm.

To evaluate the performance of the algorithm better, 80 MC runs were adopted in the experiments below. The objective function value for each iteration in ML is presented in Figure 5. Further, the number of applied data points in ML is determined by the RLS, which has already been discussed above in detail. The estimation converges well within five iterations, and the ML initial objective value is zero because the initial $m_{k}^{b^{\prime }}$ is determined by Equation (4) and $\lambda_{\mathrm{init}}=0$.

Moreover, the normal error metrics [15] were adopted in this paper. The non-orthogonal and scale factor matrices of the three-axis magnetometer could be obtained by decomposing R = MΛ, where Λ is a diagonal matrix that makes the diagonal elements of M be all 1. Several physical error metrics are defined as follows: average scale factor error, $e_{s}=\frac{1}{3}\|diag(\Lambda^{-1}\hat{\Lambda }-I)\|\times 100\%$ (in percentage), average sensor orthogonal error, $e_{o}=\frac{180}{3\pi }\|\mathrm{vec}(\hat{M}-M)\|$ (in degree), and average hard-iron effect error, $e_{o}=\frac{1}{3}\|\hat{h}-h\|$ (in Gauss). The hatted quantities denote the final estimation. The means and standard deviations of the RLS and the ML methods are provided in Table 1. From the table, it is seen that the ML method was slightly superior to the RLS method, especially in the average sensor orthogonal error, which indicates that utilizing the unbiased ML is essential to perform the estimation. Moreover, a comparison of the execution times of the conventional LS/ML [15] and the proposed RLS/ML using MATLAB is shown in Figure 6. With the increase in the number of data samples, the variation in the execution time of the proposed algorithm was less than that of the conventional method. This result demonstrated that the proposed method effectively shortens the computational time.

### 4.2. Experimental Results

To further evaluate the performance of the proposed algorithm, calibration experiments were conducted, which are discussed in this section. The algorithm was implemented in the ADIS16488BMLZ platform (ADI Inc.), which comprised of a triaxial gyroscope, triaxial accelerometer, triaxial magnetometer, and pressure sensor. The dynamic range, sensitivity, and noise density of the magnetometer are ±2.5 Gs, 0.1 mGs/LSB, and $0.042\;\mathrm{mgauss}/\sqrt{\mathrm{Hz}}$, respectively. Further, the magnetometer signals were sampled at 246 Hz. To validate the proposed algorithm, the sensor was placed in an environment amidst multiple magnetic disturbances from sources including a laptop and a ferromagnetic stair railing, and by the current carried by the sensor, as shown in Figure 7. Moreover, the experiment was conducted by the rotation of the magnetometer around central point because any change in the magnetometer positioning could affect the local magnetometer field.

The sensor measures 3000 data points for the experiments. Good calibration results were achieved when the magnetometer rotates fully in all directions, because it carries sufficient information about the ellipsoid surface. However, if the measured data do not sufficiently cover most of the ellipsoid surface, the calibration results may be poor. The proposed algorithm effectively addresses this problem. Thus, to demonstrate the performance of the proposed algorithm better, the measured data points were distributed on only a little part of the surface of the ellipsoid in this experiment, and not on most of the surface, as shown in Figure 8.

The real-time state vector estimation results and the parameters of the covariance matrix are shown in Figure 9 and Figure 10, respectively. The final number of data points was set as 1835, according to the above-mentioned condition. Moreover, the state vector and the covariance matrix tended to be stable after 1835 iterations, which agreed well with the simulation results above.

After obtaining the optimal parameters by applying the RLS method, the calibrated magnitude results could be obtained using Equation (4). Figure 11 plots the normalized magnitude result of the measured data, and the magnitude results of LS, LS/ML, and the proposed RLS and RLS/ML. It is clearly seen that all of the algorithms performed the calibration well because RLS and LS all could provide good initial estimation. In fact, it is usual that a good initial estimation could be obtained when the magnetic field was not destroyed heavily. In this test, even if only 1835 samples were required to perform the calibration, it was important to analyze the accuracy of the calibrated data in the whole 3000 samples. The magnitude results utilizing all of the 3000 points by RLS/ML was plotted in Figure 12. It is obvious that the range of the magnitudes was between 0.98 and 1.02, which also met the calibration requirement perfectly. Further, the corrected sphere fitting result by the proposed RLS/ML algorithm was plotted in Figure 13.

Finally, to compare the accuracy of the algorithms, the parameters of the reference method were utilized as the standard values. The reference calibrated parameters were obtained by utilizing all the data points, covering most of the ellipsoid surface in the same test environment. The LS/ML method was adopted to obtain the reference parameters, which has already been demonstrated to exhibit a reliable performance [15]. The reference calibrated parameters in the test are as follows:(21)$$\begin{array}{ccc} R_{\mathrm{ref}} & = & \left[\begin{array}{ccc} 2.3415 & -0.0053 & -0.0259\\ 0 & 2.3613 & -0.0255\\ 0 & 0 & 2.3938 \end{array}\right] \end{array}\;\begin{array}{ccc} h_{\mathrm{ref}} & = & \left[\begin{array}{c} 0.0165\\ 0.0245\\ 0.2061 \end{array}\right] \end{array}.$$

In order to evaluate the proposed algorithm sufficiently and reasonably, 20 experiments were conducted in many locations, including stairs, laboratory, open area, and so on. Table 2 lists the means and standard deviations of the 20 experiments by applying different algorithms. A decrease in the mean of all the error metrics could be seen from the table, especially of the average sensor orthogonal error, after applying the proposed RLS/ML method. It is clearly seen that the standard deviation of the error metrics was a little large generally. The reason for the situation was that the accuracy of the calibrated data varied greatly in different locations. In other words, the accuracy of the calibration is associated with the degree of magnetic field disturbance. For example, the calibration results of flat land were better than the stair surrounded by the ferromagnetic material. However, the maximum three error metrics ($e_{s},\;e_{o},\;\mathrm{and}\;e_{h}$) by applying RLS/ML in the 20 experiments are 0.0657, 1.3860, and 0.0242 respectively, which were all acceptable.

Moreover, RLS/ML was slightly better than LS/ML in the table, not obvious, because RLS and LS all could provide good initial estimation in most experiments. In this situation, the proposed RLS/ML could effectively shorten the computational time due to the iterative operations of RLS, as is presented in Figure 6. This was also one of the main superiority of the proposed algorithm. For example, in the test above, the LS/ML performing the calibration including 3000 samples needed 282.1227 s, while the RLS/ML only needed 46.1559 s.

## 5. Discussion

Based on the analysis of the simulation and experimental results above, the proposed RLS/ML could perform the magnetometer calibration in many different locations surrounded by magnetic field disturbance. Compared with the traditional LS algorithm, the advantages of the proposed RLS/ML can be listed as:①Shortens the computational time due to the iterative operations of RLS;②Detects the calibration implementation adaptively by the parameter s in Equation (10);③Improves the accuracy of the calibration by utilizing ML algorithm.

Although LS/ML was almost equal to RLS/ML in the accuracy, RLS/ML could effectively shorten the computational time and save memory space, which is beneficial to the engineering application. This is also one of the main advantages of the proposed algorithm.

## 6. Conclusions

A real-time magnetometer calibration algorithm based on the RLS/ML method was proposed in this paper. First, a simplified model was derived from the measured error model of the magnetometer, and an observation equation was constructed by transformation. Then, real-time parameter estimation was performed using the proposed algorithm. Finally, numerous simulations and experiments were conducted to demonstrate the efficiency of the proposed algorithm. The proposed method was not only more accurate than the conventional batch-processing method, but also reduced greatly the computational time. Moreover, the estimation results were detected in real time by state covariance matrix analysis, which improved the flexibility of the system. Due to the simplicity of the proposed measurement model of the three-axis sensor, the proposed method could be adopted for calibrating other three-axis sensors.

## Figures and Tables

**Figure 1 sensors-20-00535-f001:**
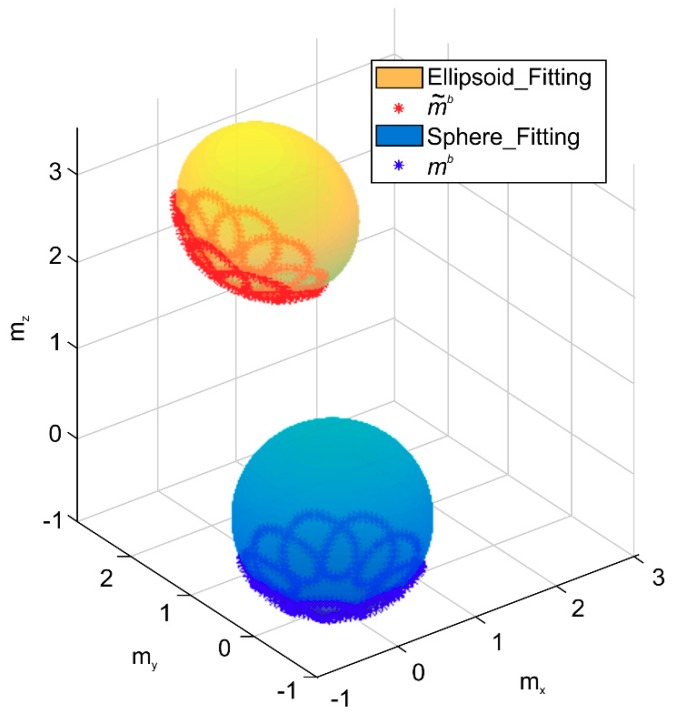
Ellipsoid and sphere fitting results.

**Figure 2 sensors-20-00535-f002:**
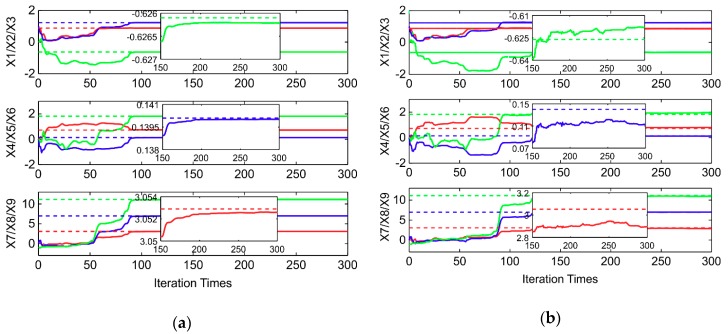
(**a**) Changes in state vector x without noise and (**b**) changes in state vector x with noise.

**Figure 3 sensors-20-00535-f003:**
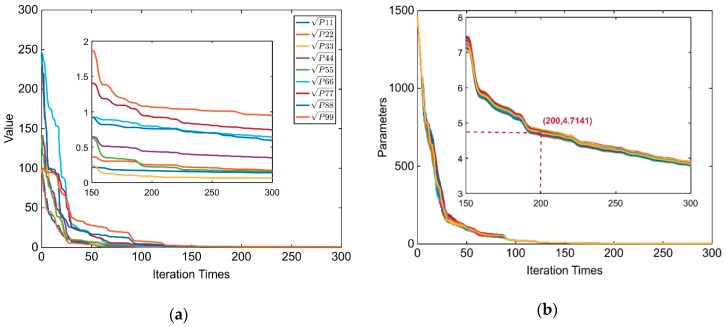
(**a**) Parameters of the covariance matrix and (**b**) result of 80 Monte Carlo simulations.

**Figure 4 sensors-20-00535-f004:**
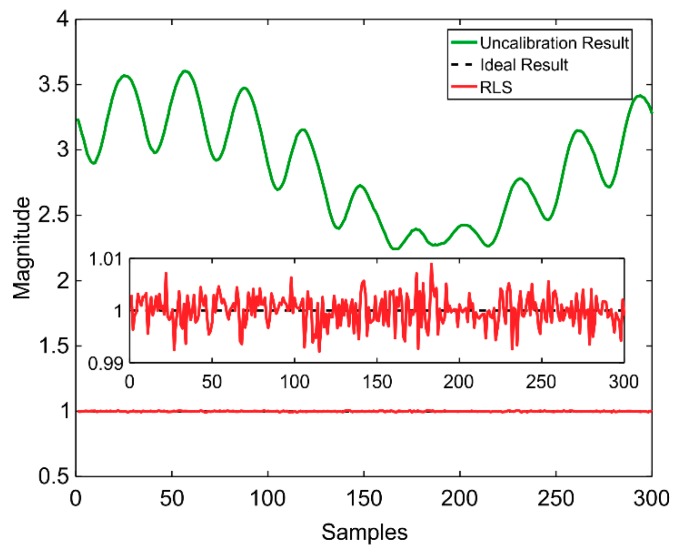
Magnitudes of the data points.

**Figure 5 sensors-20-00535-f005:**
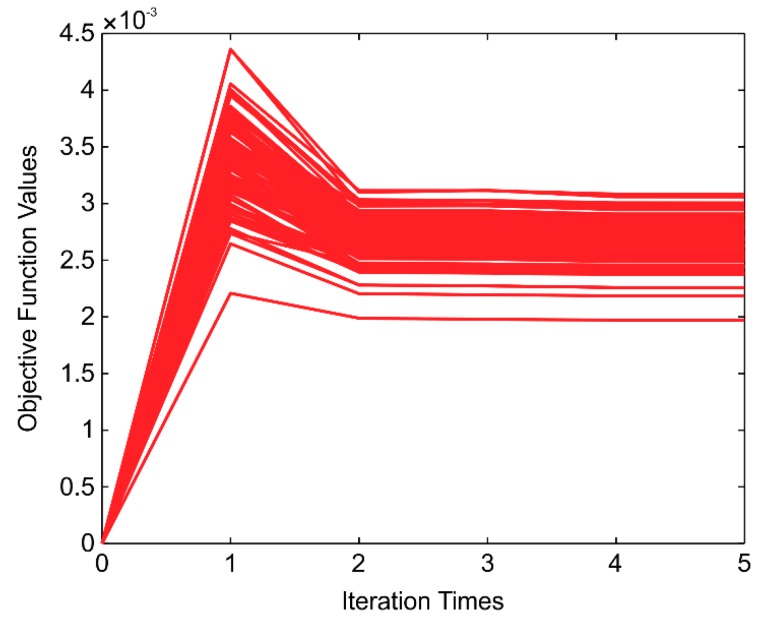
Objective function value for each iteration across 80 Monte Carlo (MC) simulations.

**Figure 6 sensors-20-00535-f006:**
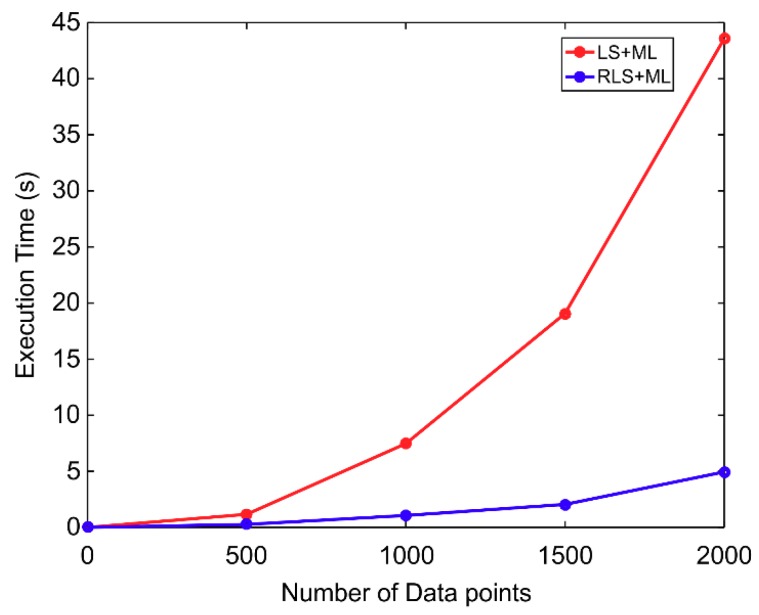
Comparison of execution times of the conventional and proposed methods.

**Figure 7 sensors-20-00535-f007:**
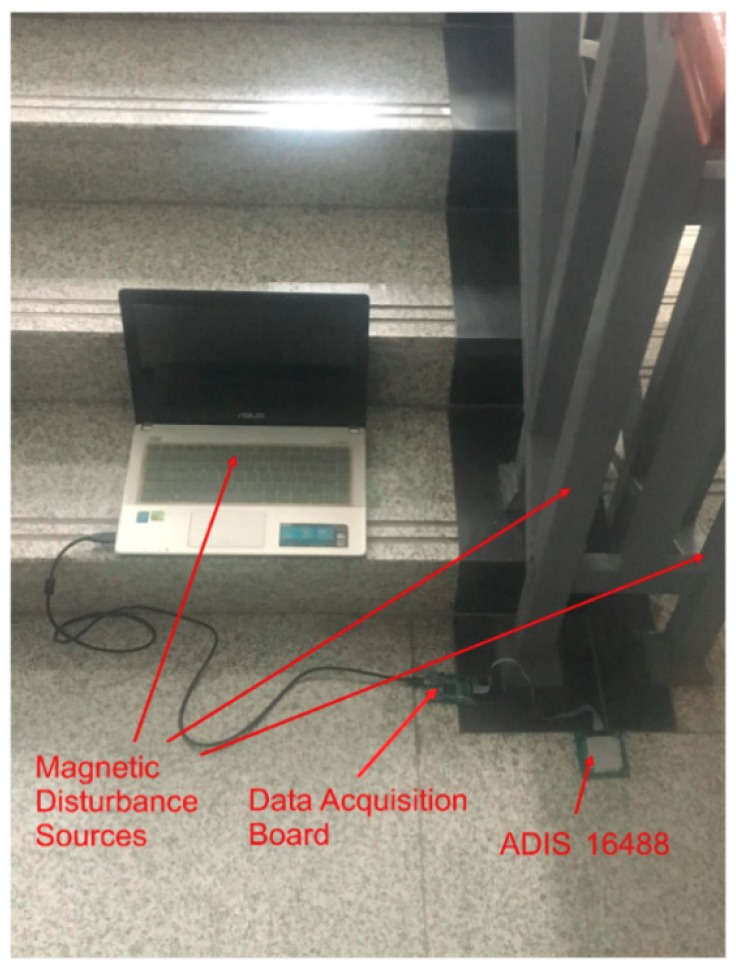
Devices used to include magnetic disturbance.

**Figure 8 sensors-20-00535-f008:**
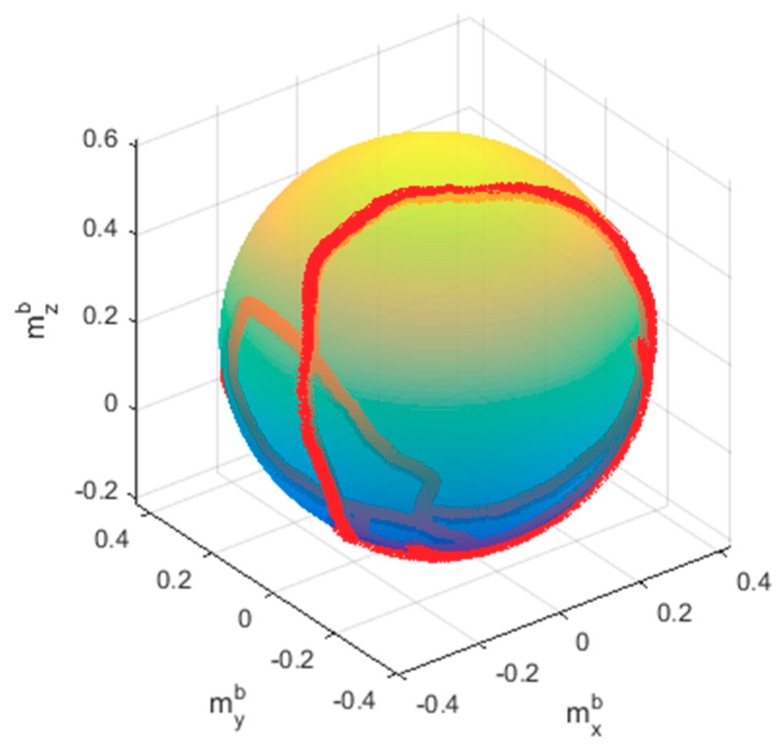
Ellipsoid fitting result.

**Figure 9 sensors-20-00535-f009:**
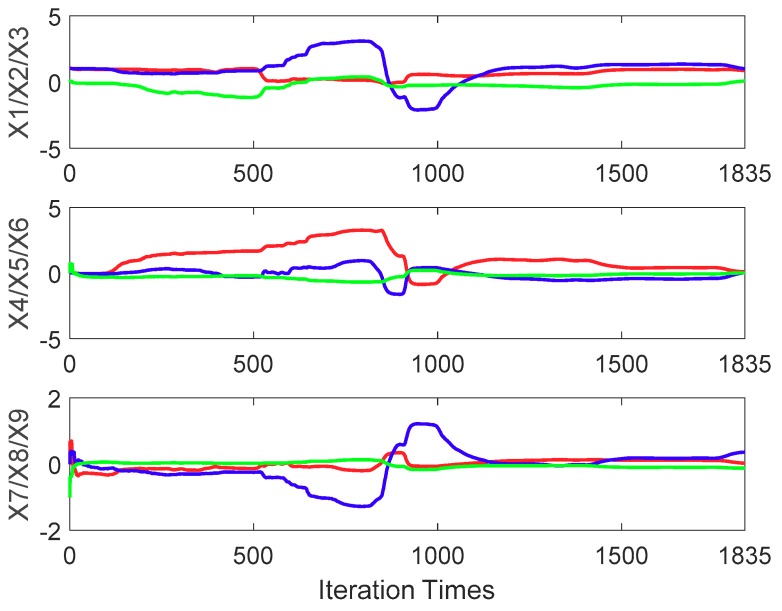
Changes in state vector x.

**Figure 10 sensors-20-00535-f010:**
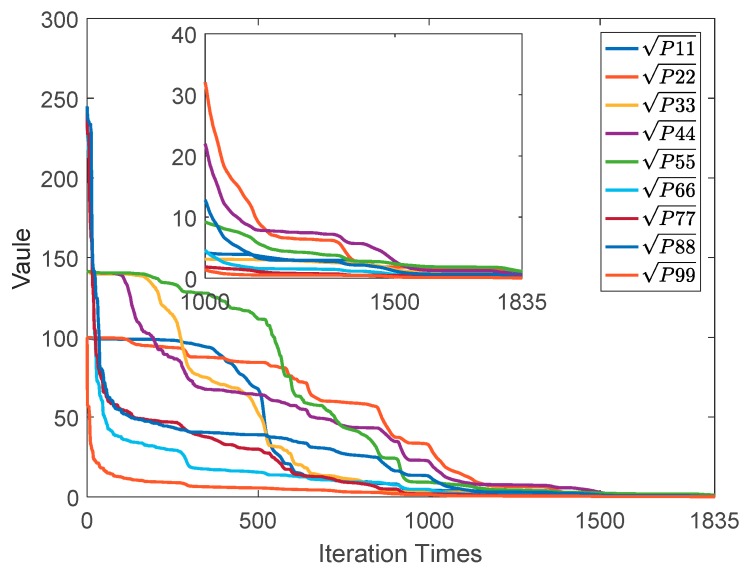
Parameters of the covariance matrix.

**Figure 11 sensors-20-00535-f011:**
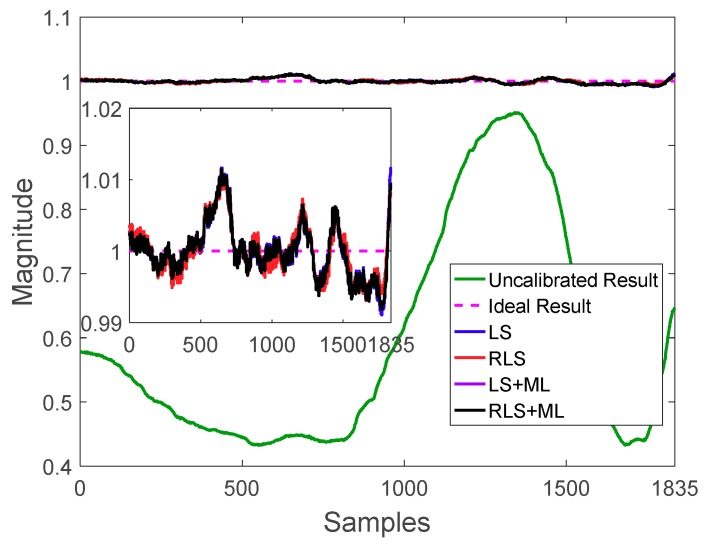
Magnitude of the data points.

**Figure 12 sensors-20-00535-f012:**
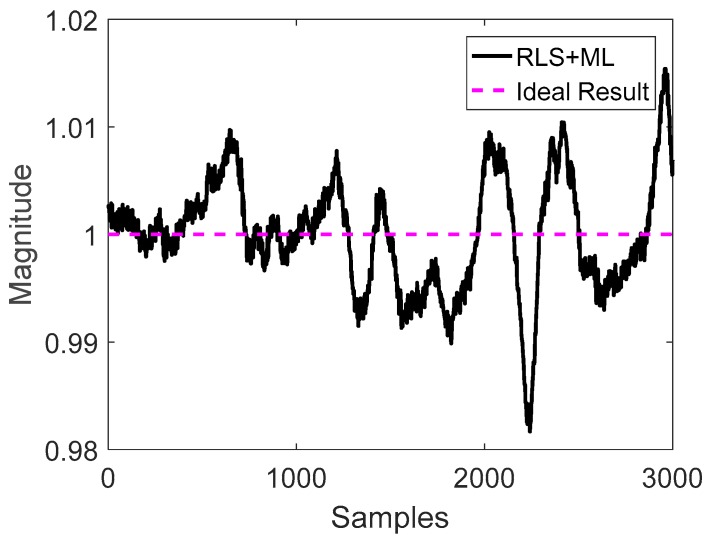
Magnitude of the whole 3000 data points by recursive least square/maximum likelihood (RLS/ML).

**Figure 13 sensors-20-00535-f013:**
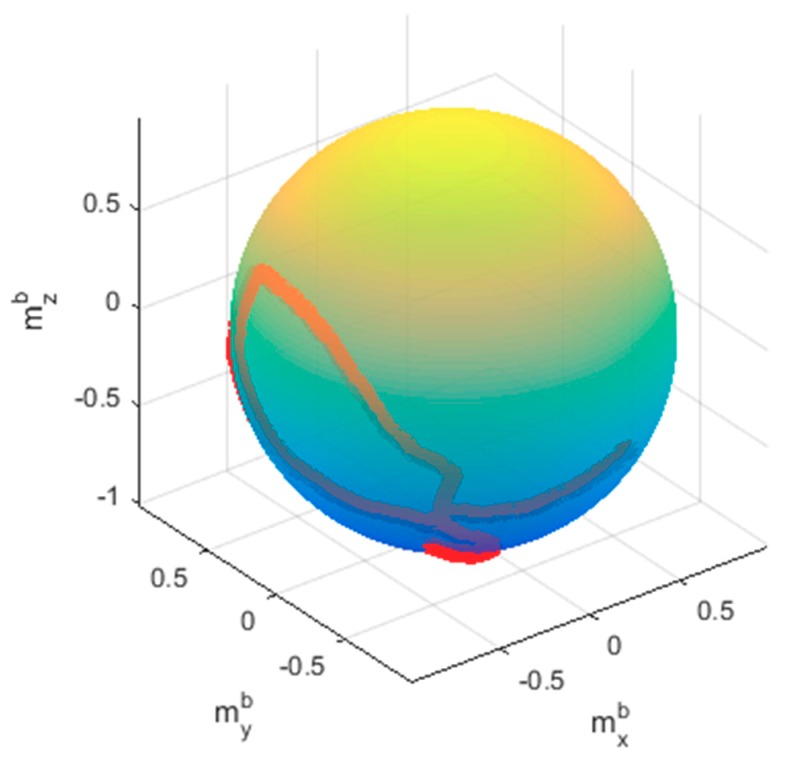
The corrected sphere fitting result.

**Table 1 sensors-20-00535-t001:** Mean (standard deviation) of three error metrics.

Methods	$e_{s}\;\left(\%\right)$	$e_{o}\;\left(\deg\right)$	$e_{h}\;(\mathrm{Gauss})$
RLS	0.0119 (0.0053)	0.4845 (0.2127)	0.0093 (0.0039)
RLS + ML	0.0047 (0.0030)	0.2035 (0.1140)	0.0040 (0.0023)

**Table 2 sensors-20-00535-t002:** Mean (standard deviation) of three error metrics in the 20 experiments.

Methods	$e_{s}\;(\%)\;$	$e_{o}\;(\deg)$	$e_{h}\;(\mathrm{Gauss})$
LS	0.0388 (0.0402)	0.9062 (0.7369)	0.0134 (0.0134)
RLS	0.0269 (0.0244)	0.8019 (0.5461)	0.0115 (0.0098)
LS + ML	0.0310 (0.0301)	0.7314 (0.5830)	0.0111 (0.0105)
RLS + ML	0.0309 (0.0300)	0.7280 (0.5803)	0.0111 (0.0105)

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
