# Peer review of "Real-Time Calibration of Magnetometers Using the RLS/ML Algorithm"

_sensors, 2020, doi:10.3390/s20020535_

Round 1

Reviewer 1 Report

The proposed magnetometer algorithm is well presented and the results are quite satisfactory.

Please proofread your manuscript again. Some serious mistakes i found were:

line 31: "accelerators" is used instead of "accelerometers"

line 37: "...magnetometers with magnetic sensors than those without magnetic sensors, ..." needs rephrasing

line 121: "bath" is used instead of "batch" and "cannot" is used instead of "can"

Reviewer 2 Report

General overview:

The presented work offers a possible new solution for a real-time calibration of three-axis magnetometers. The authors developed the algorithm starting from a model of the magnetometer output and combining in a specific way different existing algorithms.

The method developed is promising, and the result in simulated data confirmed that. The major limitation is that the result reported on real data acquisition refers only to one test lasting around 10 seconds. The author should extend the real data analysis, including different tests. The tests should be conducted in different locations and should last longer. Moreover, a real gold standard is missing in real data results (Figure 11). In fact, the so-named ‘ideal result’ should be equal to 1 only in case of no external magnetic distortion; to give only an example: if the geomagnetic field is corrupted in a different way in different positioning of the sensor, the even the perfectly calibrated magnetometer is not constant when it is moving around the space.

A major revision is required where should be included:

The issue related to the gold standard in real data acquisition More (and longer) real data acquisitions How the issue related to external magnetic distortion could be overcome (many articles are available: ‘Magnetometer Calibration and Field Mapping through Thin Plate Splines’, ‘Modeling and interpolation of the ambient magnetic field by Gaussian processes’ ) The innovative and improved contribution compared to the state of the art is better highlighted

Major:

Line 104:

The authors state that the local geomagnetic field vector has a constant magnitude. Should be underlined that the magnitude can be considered constant only if the position of the magnetometer in the navigational frame does not change over time.

This is an important issue, especially in case of external magnetic disturbances, as in the test conducted for real data acquisition.

Another clarification must be done on the sentences: ‘mÌ…n is the unit vector obtained by normalizing mn’. So, I understand that every frame of acquisition has norm equal to 1 after the normalization. What should be done if during the acquisitions the position of the magnetometer changes and consequently the value of the earth's magnetic field also changes?

The proposed method does not take into account this issue.

Have hypotheses been made about the direction of the magnetic field? For the resolution of the calibration algorithm, is it important to know the components of the terrestrial magnetic field vector? This point should be clarified.

Line 169 – Section ‘simulation results’

Has the complete identifiability of the parameters been verified?

Line 234 – Section ‘experimental design’

It was considered that in an environment amidst multiple magnetic disturbances, any change in the magnetometer positioning can affect the local magnetic field? It should be underlined.

The major limitation of the presented study is that the results reported on real data acquisition refers only to one test lasting around 10 seconds. The author should extend the real data analysis, including different tests. The tests should be conducted in different locations and should last longer.

245: Only 3000 data points were acquired? It is around 12 seconds (frequency of 246 Hz). The stability of the calibration parameter should be verified on a longer acquisition.

Line 276 – Figure 11:

Is there an explanation why the values representing the LS+ML results is so far from 1? They are at least 20 times worse than the others.

Why does the x axis end at 1200 frame (1196 I suppose)? What are the results of the remaining 1800 frames?

Line 286: Section ‘conclusion’

The discussion section is missing, and the short conclusion presented can not replace it.

Minor:

Line 29: Using only MIMU without external references an accurate orientation system can be obtained (there are no robust method for positioning estimation).

Line 31: accelerometer (not acceleretors)

Line 30: The magnetometer is not an inertial sensor; so I suggest to use Magnetic and Inertial Measurement Units (MIMU), instead of Micro Inertial Measurement Units

Line 179: The sentence: ‘Figure 2, which plots the changes in the state vector ? with or without noise’ should be improved

Line 219: I think the multiplying factor should be 180/(2pi), not 180/(3pi)

Table 1 and 2: eh (Gauss) is missing

Round 2

Reviewer 2 Report

Some of the requested improvements were implemented, but there are still important open points.

A major revision is required:

It is not usual to show results in the discussion section. Figure 13 and its explanation must be in the results section. The results of the 20 different tests should be presented. Maybe a table with the mean error over the different tests could be presented. Referred to point 6 of previous review: Even if only 1196 are required to perform the calibration,

it is important to analyse the accuracy of the calibrated data in the last 1804 samples. In this way the calibrated parameter will be tested on a different dataset.

The experimental results must be clarified: Why e0 is around 15 degrees?

There is an explanation about that? Usually the non-orthogonal error are around 1-2 degrees. Unless there is a valid explanation, this error is not acceptable

Round 3

Reviewer 2 Report

The authors completely changed the results of the paper with respect to the first version, and this makes the results unreliable in my opinion.

In the first version of the paper the authors presented promising results (in their opinion), but in the last version the same results 'cannot be presented as one successful calibration example'.   The authors must know when a calibration result is good or not.   Moreover, 10 seconds of data acquisition are not enought.
